# Arazyme Suppresses Hepatic Steatosis and Steatohepatitis in Diet-Induced Non-Alcoholic Fatty Liver Disease-Like Mouse Model

**DOI:** 10.3390/ijms20092325

**Published:** 2019-05-10

**Authors:** Hua Li, Wonbeak Yoo, Hye-Mi Park, Soo-Youn Lim, Dong-Ha Shin, Seokho Kim, Ho-Yong Park, Tae-Sook Jeong

**Affiliations:** 1Industrial Bio-Materials Research Center, KRIBB, Daejeon 34141, Korea; leehua@kribb.re.kr (H.L.); yoow@kribb.re.kr (W.Y.); hyemi0707@nate.com (H.-M.P.); isy3929@naver.com (S.-Y.L.); kims@kribb.re.kr (S.K.); 2Insect Biotech Co., Ltd., Daejeon 34054, Korea; dhshin@insectbiotech.co.kr

**Keywords:** arazyme, diet therapy, non-alcoholic fatty liver disease, SREBP-1, steatohepatitis, steatosis

## Abstract

Arazyme, a metalloprotease from the spider *Nephila clavata*, exerts hepatoprotective activity in CCL_4_-induced acute hepatic injury. This study investigated the hepatoprotective effects in high-fat diet (HFD)-induced non-alcoholic fatty liver disease-like C57BL/6J mice. The mice were randomly divided into four groups (*n* = 10/group): the normal diet group, the HFD group, the arazyme group (HFD with 0.025% arazyme), and the milk thistle (MT) group (HFD with 0.1% MT). Dietary supplementation of arazyme for 13 weeks significantly lowered plasma triglyceride (TG) and non-esterified fatty acid levels. Suppression of HFD-induced hepatic steatosis in the arazyme group was caused by the reduced hepatic TG and total cholesterol (TC) contents. Arazyme supplementation decreased hepatic lipogenesis-related gene expression, sterol regulatory element-binding transcription protein 1 (*Srebf1)*, fatty acid synthase (*Fas*), acetyl-CoA carboxylase 1 (*Acc1*), stearoyl-CoA desaturase-1 (*Scd1*), *Scd2*, glycerol-3-phosphate acyltransferase (*Gpam*), diacylglycerol *O*-acyltransferase 1 (*Dgat1*), and *Dgat2*. Arazyme directly reduced palmitic acid (PA)-induced TG accumulation in HepG2 cells. Arazyme suppressed macrophage infiltration and tumor necrosis factor α (*Tnfa*), interleukin-1β (*Il1b*), and chemokine-ligand-2 (*Ccl2*) expression in the liver, and inhibited secretion of TNFα and expression of inflammatory mediators, *Tnfa*, *Il1b*, *Ccl2*, *Ccl3*, *Ccl4*, and *Ccl5*, in PA-induced RAW264.7 cells. Arazyme effectively protected hepatic steatosis and steatohepatitis by inhibiting SREBP-1-mediated lipid accumulation and macrophage-mediated inflammation.

## 1. Introduction

Non-alcoholic fatty liver disease (NAFLD) comprises chronic conditions including steatosis, steatohepatitis, and cirrhosis [1]. Hepatic steatosis causes increased hepatic lipid contents that mainly accumulate as triglyceride (TG) fat. During hepatic lipid accumulation, sterol regulatory element-binding protein 1 (SREBP-1) and carbohydrate responsive element-binding protein (chREBP) play critical roles as transcription factors in the synthesis of fatty acids and TG. The steatotic condition can further develop into steatohepatitis by triggering inflammatory cytokine production [2]. Non-alcoholic steatohepatitis is characterized by recruited macrophage-related inflammation and hepatic collagen synthesis-related fibrosis. The progression of NAFLD is closely associated with glucose homeostasis, insulin resistance, and inflammation: Notably, the prevalence rate of NAFLD is higher in overweight/obese patients with type 2 diabetes [3]. Furthermore, hepatic steatosis is a significant contributor to dyslipidemia, which dramatically increases the risk factors of atherogenic cardiovascular diseases [4]. Additionally, chronic inflammatory conditions such as periodontitis elevates the blood inflammatory factors and are significantly associated with NAFLD development [5]. Therefore, NAFLD is considered a notable risk factor for metabolic syndrome. 

The nutritional value of bugs, insects, and spiders is comparable to those of commonly eaten meats [6]. In addition to meat and plant-derived protein, many insects are also a good protein source for humans. More recently, insects have been identified as an alternative protein source for the Western world, and this use has been supported by organizations including the Food and Agriculture Organization of the United Nations (FAO) [7]. Globally, the most frequently consumed species of bugs are beetles, caterpillars, bees, wasps, and ants. Edible insects can be excellent sources of not only protein but also minerals, vitamins, fat, and energy [8]. Recently, a growing interest in insect-based bioactive products has been accompanied by an increasing demand for alternative protein sources. Some insect-derived natural products and their components have been found to have bioactivity, exerting antioxidant, anti-inflammatory, hepatoprotective, and anti-obesity effects [9,10,11,12]. 

The *Serratia proteamaculans* (*Aranicola proteolyticus*), a bacterium isolated from the gut of the spider *Nephila clavata*, produces a 51.5-kDa metalloprotease, known as arazyme [13]. Arazyme has been approved as a food additive (registration No. 2007-SA-0007) by the Ministry of Food and Drug Safety in the Republic of Korea. Arazyme is used as an ingredient in health-promoting functional foods, as an additives in cosmetics, and as a feed additives. A mass of arazyme can be harvested from bacterial culture. Purified arazyme exhibits biochemical and genetic characteristics of an extracellular protease [14]. Exogenous proteases are involved in several biological processes related to cancer, infectious diseases, and inflammation [15]. Previous studies have reported the effects of arazyme, including anti-atopic, anti-inflammatory, and hepatoprotective effects [16,17,18,19]. However, the hepatoprotective effects of arazyme in chronic NAFLD have not been documented. This study used C57BL/6J mice fed a high-fat diet (HFD) as an in vivo chronic NAFLD model and palmitic acid (PA)-induced HepG2 and RAW264.7 cells as in vitro NAFLD models [20]. The objective of this study was to assess the hepatoprotective activity of arazyme in HFD-induced NAFLD-like mice, HepG2 hepatocytes, and RAW264.7 macrophages and to study the molecular mechanisms underlying its effects on NAFLD.

## 2. Results

### 2.1. Effects of Arazyme on Metabolic Parameters

To investigate the activities of arazyme on metabolic parameters, 0.025% (*w*/*w*) dietary arazyme was fed along with an HFD to C57BL/6J mice. Milk thistle (0.1% *w*/*w*, *Silybum marianum*, MT) was used as a positive control. After a 13-week feeding period, the final body weight were significantly increased in each group as compare to that in the initial body weight. The HFD-fed mice exhibited characteristics of obesity in terms of body weight, weight gain, and food intake compared to those in the normal diet (ND) group at final conditions (Table 1). However, there was no change in final body weight and food intake among the HFD-fed groups. Liver weight decreased in the arazyme group by 5.1% compared to that in the HFD group, although this change was not significant. 

### 2.2. Effects of Arazyme on Plasma Biomarkers

After a 13-week feeding, final glucose and insulin levels were significantly increased in all groups compared to those in the initial levels. At the final conditions, treatment with arazyme or MT markedly reduced plasma glucose, homeostatic model assessment for insulin resistance (HOMA-IR), TG, total cholesterol (TC), and non-esterified fatty acid (NEFA) levels in HFD-fed mice (Table 2). Specifically, the arazyme and MT groups exhibited significantly reduced HFD-induced plasma lipid profiles, as well as TG (17.0% and 21.7%, respectively) and NEFA (17.2% and 18.7%, respectively) levels. Glycated hemoglobin (HbA1c) levels were reduced in the arazyme group by 4.8% compared to that in the HFD group, although this change was not significant. Plasma aspartate transaminase (AST) and alanine transaminase (ALT) levels were significantly higher in the HFD group than in the ND group (Table 2). Arazyme or MT supplementation decreased plasma AST (9.9% and 4.8%, respectively) and ALT (8.4% and 11.2%, respectively) levels compared to those in the HFD group. However, there were no changes in high-density lipoprotein cholesterol (HDL-C) or HDL-C/TC levels among the HFD-fed groups.

The HFD group showed significant glucose intolerance compared to that of the ND group after an 11-week feeding. Arazyme administration significantly improved oral glucose tolerance at 15 and 30 min after gavage with glucose (Figure 1A). The areas under the glucose-concentration–time curves revealed that supplementation of arazyme dramatically improved glucose tolerance (Figure 1B). These results indicate that arazyme effectively attenuated HFD-induced changes in metabolic parameters and liver injury.

### 2.3. Arazyme Reduced HFD-Induced Hepatic Steatosis

Next, we investigated whether arazyme affected hepatic lipid metabolism. The number of lipid droplets in the liver and the hepatic TG and TC contents were higher in the HFD group than in the ND group (Figure 2A–D). Arazyme and MT administration both markedly blocked the effects of HFD-induced hepatic steatosis. 

Next, transcription factors and target genes associated with the development of NAFLD were investigated. HFD dramatically increased MLX interacting protein like (*Mlxipl*), sterol regulatory element-binding transcription protein 1 (*Srebf1*), and *Srebf2* mRNA levels and enhanced the expression of lipogenic target genes in the liver (Figure 2A,B). In the arazyme group, the expression levels of *Srebf1* were significantly decreased, while those of the transcription factors *Mlxipl* and *Srebf2* were unaffected (Figure 3A). The target genes of SREBP-1, including fatty acid synthase (*Fas*), acetyl-CoA carboxylase 1 (*Acc1*), stearoyl-CoA desaturase-1 (*Scd1*), *Scd2*, glycerol-3-phosphate acyltransferase (*Gpam*), diacylglycerol *O*-acyltransferase 1 (*Dgat1*), and *Dgat2*, were significantly reduced in arazyme-treated mice (Figure 3A–C). However, arazyme did not affect the mRNA levels of lipid utilization-related factors, including peroxisome proliferator-activated receptor alpha (*Ppara*), acyl-coenzyme A oxidase (*Acox*), carnitine palmitoyltransferase alpha (*Cpt1a*), and uncoupling protein 2 (*Ucp2*) (Figure 3D). Moreover, SREBP-1 and FAS protein expression levels were markedly reduced following arazyme supplementation (Figure 3E,F). In addition, MT administration reduced the expression of HFD-induced lipid accumulation-related transcription factors and their target genes, including *Mlxipl, Srebf1*, *Srebf2*, *Fas*, *Acc2*, *Scd1*, *Scd2*, *Gpam*, *Dgat1*, and *Dgat2*. Thus, treatment with arazyme improved HFD-induced hepatic steatosis by inhibiting the expression of lipogenic genes and proteins. Although arazyme and MT resulted in similar anti-lipogenic effects in HFD-fed mice, the effective target points of arazyme and MT were slightly different. 

### 2.4. Arazyme Inhibited Lipid Accumulation in HepG2 Cells

Next, the anti-lipogenic effect of arazyme was investigated in HepG2 hepatocytes. Arazyme did not exhibit cytotoxicity at concentrations of up to 0.1 μg/mL in HepG2 cells (Figure 4A,B). Treatment with arazyme significantly reduced PA-induced increases in the intracellular TG content in a dose-dependent manner (Figure 4C,D). Additionally, arazyme markedly inhibited the protein expression of SREBP-1 and FAS in PA-induced HepG2 cells (Figure 4E,F). Thus, arazyme directly suppressed hepatic steatosis via an SREBP-1-related pathway.

### 2.5. Arazyme Suppressed Steatohepatitis and Macrophage Recruitment

Chronic hepatic steatosis can develop into steatohepatitis accompanied by inflammation through the recruitment of macrophages. After 13 weeks of a HFD, macrophages were recruited to the liver. However, arazyme or MT supplementation reduced F4/80 positive macrophage recruitment compared with that in the HFD group (Figure 5A). Moreover, treatment with arazyme significantly reduced tumor necrosis factor α (*Tnfa*), interleukin-1β (*Il1b*), and chemokine ligand 2 (*Ccl2*) mRNA levels compared with those in the HFD group (Figure 5B). The anti-inflammatory effects were more pronounced in the arazyme group than in the MT group.

### 2.6. Arazyme Inhibited PA-Induced Inflammation in RAW264.7 Macrophages

The effect of arazyme on inflammatory conditions was investigated in PA-induced RAW264.7 macrophages. Arazyme did not exhibit cytotoxicity at concentrations of up to 0.5 μg/mL in RAW264.7 cells (Figure 6A,B). Treatment with arazyme significantly reduced PA-induced increases in TNFα secretion and the mRNA expression of inflammatory mediators, including *Tnfa*, *Il1b*, *Ccl2*, *Ccl3*, *Ccl4*, and *Ccl5* in a dose-dependent manner (Figure 6C,D). Thus, arazyme may suppress the recruitment and activation of macrophages by inhibiting the expression of inflammatory chemokines and cytokines.

## 3. Discussion

In this study, we found that arazyme exerts hepatoprotective effects by inhibiting hepatic steatosis and steatohepatitis in mice with chronic HFD-induced NAFLD. Previous studies have found that arazyme suppresses inflammatory mediators including reactive oxygen species, monocyte chemoattractant protein 1 (MCP-1), interleukin-6, and vascular cell adhesion molecule-1 in lipopolysaccharide (LPS)-induced human umbilical vein endothelial cells [16]. In addition, arazyme has shown anti-atopic effects through the downregulation of inflammatory mediators and by enhancing the skin barrier in atopic dermatitis-related cell and mouse models [17,18]. Furthermore, arazyme restores acute hepatic injury by activating the SMP30 (TGF-β)/Smad pathway and increasing the expression of antioxidant mediators in CCl_4_-induced SMP30-knockout mice [19]. 

Hepatic steatosis in NAFLD may be caused by several processes, including an increased influx of free fatty acids stemming from overnutrition, increased hepatic de novo lipogenesis, increased insulin resistance, reduced fat oxidation, and decreased secretion of very low-density lipoprotein [20]. Overnutrition promotes the de novo synthesis of free fatty acids and TG. The transcription factors SREBP-1, SREBP-2, and chREBP play critical roles in de novo lipogenesis: SREBP-1 regulates the fatty acid- and TG synthesis-related gene expressions of genes involved in fatty acid and TG synthesis, while SREBP-2 regulates those involved in cholesterol synthesis and uptake [21,22] and chREBP regulates those involved in glycolysis, gluconeogenesis, and lipogenesis [23,24]. SREBP-1 is synthesized from precursors in the endoplasmic reticulum (ER) [25]. SREBP-cleavage-activating protein drives shuttling of the precursor SREBP-1 to be shuttled to the Golgi, where it is proteolytically cleaved by site-1 and site-2 proteases. Upon release from the ER, mature SREBP-1 translocates to the nucleus, where it binds to sterol regulatory elements in the promoters of target genes [26]. Once bound, SREBP-1 is phosphorylated and facilitates the recruitment of coactivators to promoters to stimulate the transcription of target genes, including *FAS*, *ACCs*, and *SCDs* [27]. Moreover, overexpression of SREBP-1 causes insulin resistance by inhibiting AKT translocation to membrane and phosphorylation [28]. In terms of the expression of de novo lipogenesis-associated transcription factors, arazyme treatment exerted a different effect than MT, which was used as a positive control. Arazyme supplementation decreased the gene expression of *Srebf1*, but not *Srebf2* or *Mlxipl*. This finding may be related to the fact that arazyme treatment caused a more significant reduction in TG than in TC according to plasma lipid profiles. Arazyme treatment also led to decreases in mature SREBP-1 and FAS protein levels, as well as SREBP-1 target gene mRNA levels, including *Fas*, *Acc1*, *Scd1*, *Scd2*, *Gpam*, *Dgat1*, and *Dgat2*, inhibiting hepatic fatty acid and TG synthesis. These anti-lipogenic effects of arazyme were also shown to directly reduce PA-induced intracellular lipid accumulation in HepG2 hepatocytes. 

Obesity is commonly considered to be closely related to hepatic steatosis in NAFLD [4]. Hepatic steatosis can develop into steatohepatitis when associated with hepatocyte injury, pro-inflammatory mediators, and fibrosis [29]. During progression of NAFLD, inflammation is considered as a “second hit” using the “two-hit” hypothesis [3]. Resident Kupffer cells, self-renewing embryo-derived local macrophages, and recruited macrophages play critical roles in the production of inflammatory mediators in steatohepatitis [30]. Kupffer cells can promote the steatohepatitis through recruitment the leukocytes by activation of intracellular adhesion molecule 1 (ICAM-1) and vascular cellular adhesion molecule 1 (VCAM-1) [31]. Kupffer cells and macrophages derived by monocytes produce a potent chemoattractive mediator, MCP-1. MCP-1 interacts with its receptor, C-C motif chemokine receptor 2 (CCR2), to recruit more macrophages, driving the progression of steatohepatitis [32]. Activated macrophages release inflammatory cytokines and chemokines, including TNFα, IL-1β, MCP-1, CCLs, and adhesion molecules, to promote the development of inflammation and fibrosis [31]. Increased cytokines, especially TNFα generates inflammation and hepatocyte cell death. TNFα also is first defined cytokine in NAFLD and induces insulin resistance by activation of NF-κB signaling [3]. Inflammation promotes the wound healing process that leads to collagen deposition and fibrosis. We found that arazyme suppressed steatohepatitis and inhibited the expression of macrophage marker F4/80 and the inflammatory cytokines *Tnfa*, *Il1b*, and *Ccl2* in the liver. Adipose tissue secretes inflammatory adipokines, which also promote liver injury and steatohepatitis [33]. These anti-inflammatory effects of arazyme were also confirmed in PA-induced RAW264.7 macrophages. These results indicate that arazyme inhibits the inflammation by suppressing TNFα secretion and the gene expression of inflammatory mediators in vivo and in vitro.

Chronic steatohepatitis can be paralleled by an increase in collagen fibers produced by hepatic stellate cells. Inflammatory chemokines and cytokines produced by resident Kupffer cells or recruited macrophages can induce the hepatic stellate cells to transdifferentiate into myofibroblasts [34]. The recruited macrophages promote hepatic fibrosis through the activation of NF-κB in hepatic stellate cells [35]. Resident Kupffer cells also play important roles in hepatic fibrosis through the activation TGF-β-dependent signaling in hepatic stellate cells [34]. As part of the pool of extracellular matrix-producing cells, myofibroblasts act as critical mediators of collagen synthesis and the development of hepatic fibrosis. It has been demonstrated that HFD-fed mice develop chronic hepatitis characterized by inflammation and fibrosis, accompanied by the upregulation of collagen protein and mRNA levels [36]. However, macrophage-expressed matrix metalloproteases (MMP) can reverse hepatic injury and fibrosis by matrix degradation [34]. Pereira et al. reported that the metalloprotease arazyme cross-reacted with antibodies against MMP-8 to suppress metastatic murine melanoma [37]. MMP-8, also referred to as collagenase-2, is considered to play an essential role in the development of hepatitis in TNF-induced conditions [38]. MMP-8 not only functions in collagen degradation but also induces hepatic stellate cells to differentiate into myofibroblasts upon inflammatory stimuli [39]. The metalloprotease arazyme may contribute to suppression of myofibroblast differentiation and progression of inflammation-related fibrosis, preventing further development of NAFLD. However, further studies are needed to clarify whether arazyme acts as a direct collagenase or competes with the substrates of a metalloprotease involved in fibrosis development.

In summary, this study investigated the underlying mechanisms of the hepatoprotective effects of arazyme in HFD-induced chronic NAFLD-like mice. Arazyme reversed HFD-induced NAFLD-associated phenotypes, including glucose homeostasis-related factors and lipid profiles. These effects appear to result from the suppression of SREBP-1-mediated hepatic lipogenesis-related genes, including *Fas*, *Acc1*, *Scd1*, *Scd2*, *Gpam*, *Dgat1*, and *Dgat2*. Arazyme also decreased PA-induced lipogenesis by inhibiting SREBP-1 and FAS expression in HepG2 hepatocytes directly. Arazyme decreased macrophage recruitment by inhibiting the expression of inflammatory cytokines, *Tnfa*, *Il1b*, and *Ccl2*, in the liver. Additionally, arazyme suppressed TNFα secretion and the expression of chemokine- and cytokine-related genes, including *Tnfa*, *Il1b*, *Ccl2*, *Ccl3*, *Ccl4*, and *Ccl5* in PA-induced RAW264.7 macrophages. Thus, arazyme protects against hepatic steatosis and hepatitis and acts to inhibit further progression of NAFLD. Taken together, these results indicate that arazyme may be an excellent pharmacological candidate for the treatment of chronic NAFLD. 

## 4. Materials and Methods

### 4.1. Animals and Diets

All mice used in this study were 4-week-old male C57BL/6J mice purchased from the Laboratory Animal Resource Center of the Korea Research Institute of Bioscience and Biotechnology (KRIBB, Daejeon, Korea). Mice were housed in a facility under a controlled temperature (22 ± 2 °C), humidity (50 ± 5%), and lighting (12-h light/dark cycle) with free access to gamma-irradiated food and autoclaved water in a specific pathogen-free facility at KRIBB. During the 3-week acclimation period, mice were fed a 10 kcal% standard rodent diet (Research Diets, Inc., New Brunswick, NJ, USA). The mice were randomly divided into four groups (*n* = 10 per group): the ND group, which was fed a 10 kcal% normal diet; the HFD group, which was fed a 60 kcal% diet (Research Diets, Inc.) with no supplementation; the arazyme group, which was fed an HFD with 0.025% (*w*/*w*) arazyme; and the MT group, which was fed an HFD with 0.1% (*w*/*w*) MT seed extract for 13 weeks. Milk thistle (*Silybum marianum*) was used as a positive control, as it is widely used as an herbal functional food and is known to prevent liver injury from various stimuli, including in HFD-induced NAFLD [40,41]. All animal experiments were approved by the Animal Care and Use Committee of KRIBB (KRIBB-AEC-16050, dated March 18, 2016) and performed in accordance with the Animal Research Reporting In Vivo Experiment (ARRIVE)-based guidelines at KRIBB.

### 4.2. Measurement of Metabolic Parameters

After 11 weeks of supplementation, mice were subjected to an oral glucose tolerance test (OGTT) as previously reported [42]. At the end of the experiment, mice were sacrificed after a 12-h fast, and samples were collected from all mice. The blood from the inferior vena cava was collected in heparin-coated tubes. Plasma metabolic parameters were analyzed as previously reported [43]. 

### 4.3. Histological Analysis of the Liver

Fresh liver tissues were fixed in 10% formalin, embedded in paraffin, and cut into 4-µm sections for hematoxylin-eosin (H&E) staining. To measure macrophage recruitment, tissue sections were retrieved antigen using a heat-induced EDTA solution, and F4/80 antibody (Abcam, Cambridge, UK) was used for immunohistochemistry (IHC). F4/80 IHC was conducted with the ABC kit (Vector Laboratories, Inc., Burlingame, CA, USA) and a DAB substrate kit (Vector Laboratories, Inc.) according to the manufacturer’s recommendations. Cryostat sections were fixed in 10% formalin and used for Oil-red O staining. Images of stained liver sections were obtained with an Olympus microscope-camera system (Tokyo, Japan).

### 4.4. Real-Time Quantitative RT-PCR (qRT-PCR)

Total RNA was extracted using TRI reagent solution (Ambion, Carlsbad, CA, USA) and an RNeasy mini kit (Qiagen, Valencia, CA, USA). cDNA was synthesized from total RNA using the Omniscript cDNA synthesis kit (Qiagen). RNA expression was quantified by real-time qRT-PCR using cDNA and SYBR Green Master Mix (Roche Diagnostics, Mannheim, Germany) on a 7500 Real-Time PCR system (Applied Biosystems, Foster City, CA, USA). The primer sequences are listed in Table 3. 

### 4.5. Western Blotting Analysis

The proteins from liver and cell extracts were subjected to Western blotting analysis with antibodies against FAS and SREBP-1. Protein expression levels were determined by an enhanced-chemiluminescence Western blotting kit (Thermo Fisher Scientific Inc., Waltham, MA, USA) and LAS-4000 luminescent-image reader (Fuji Photo Film, Tokyo, Japan). The band intensities were analyzed using Fujifilm Image MultiGauge (Fuji Photo Film).

### 4.6. Experiments in HepG2 Cells and RAW264.7 Cells

The HepG2 human hepatocellular carcinoma cell line and RAW264.7 murine macrophage cell line were obtained from the American Type Culture Collection (Manassas, VA, USA). HepG2 and RAW264.7 cells were cultured as previously reported [42,43]. 

The 0.1 mM PA-inducing medium was prepared with serum-free DMEM using the methods described by Kim et al. [44]. PA-induced intracellular lipid accumulation was assessed by Oil-red O staining and the measurement of TG contents after 24-h treatment. The TG contents were measured using a commercially available kit (Asan Pharm. Co., Seoul, Korea). 

RAW264.7 cells were plated at 1.5 × 10^7^ cells per well in 6-well plates and incubated with 0.2 mM PA with arazyme for 2 h. The medium was used to determine the TNFα secretion with a TNFα ELISA kit (KOMA Biotech, Seoul, Korea). The cells were harvested, and total RNA was extracted using TRI reagent solution (Ambion). 

### 4.7. Statistical Analyses

Results from the animal experiments are shown as the means ± standard errors (SE). Statistical differences were analyzed by one-way or two-way ANOVA with Tukey’s post-hoc test using JMP^®^ software (SAS Institute, Cary, NC, USA). Results from cell experiments are presented as the means ± standard deviations (SD) and were analyzed using Student’s *t*-tests to identify significant differences between the two groups. A *p* value < 0.05 was considered statistically significant.

## Figures and Tables

**Figure 1 ijms-20-02325-f001:**
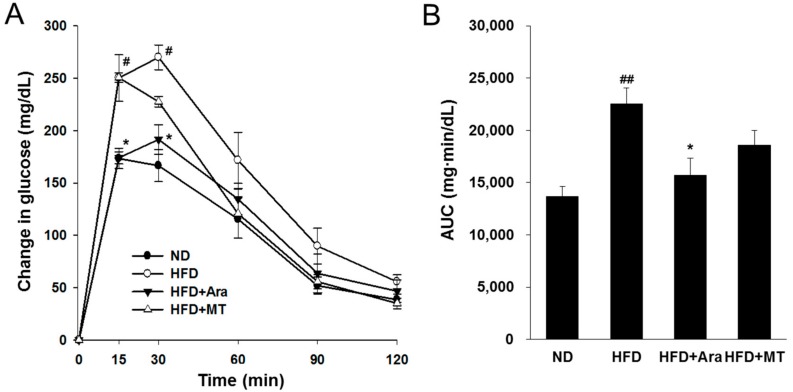
Arazyme improved oral glucose tolerance test (OGTT) in HFD-fed NAFLD-like mice. (**A**) Blood glucose levels were measured at 0, 15, 30, 60, 90, and 120 min after glucose (2 g∙kg^−1^ body weight) administration. ^#^
*p* < 0.05 vs. ND; * *p* < 0.05 vs. HFD by two-way ANOVA. (**B**) The area under the curve (AUC) of plasma glucose during OGTT. Data are presented as means ± SE (*n* = 4). ^##^
*p* < 0.01 vs. ND; * *p* < 0.05 vs. HFD by one-way ANOVA.

**Figure 2 ijms-20-02325-f002:**
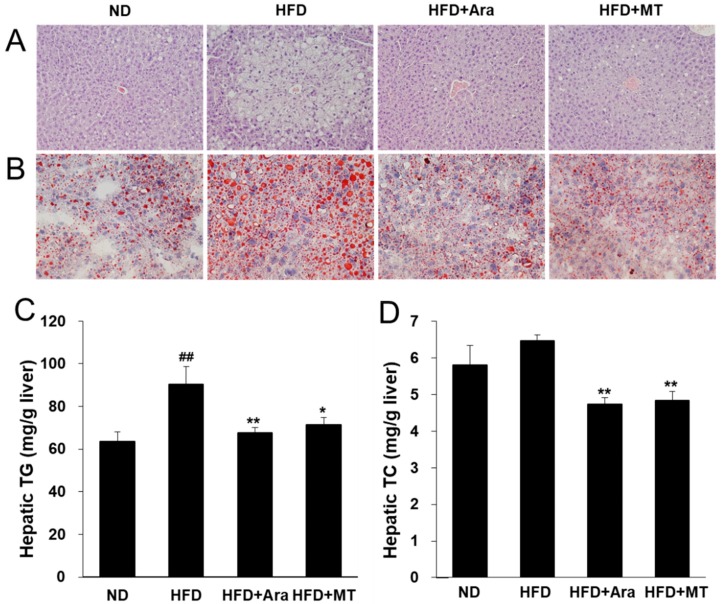
Arazyme reduced hepatic steatosis in HFD-fed NAFLD-like mice. Histology of the livers stained with hematoxylin-eosin (H&E) (**A**) and Oil-red O (**B**) (× 200 magnification). (**C**,**D**) Hepatic TG and TC contents. Data are presented as means ± SE (*n* = 10). ^##^
*p* < 0.01 vs. ND; * *p* < 0.05, ** *p* < 0.01 vs. HFD by one-way ANOVA.

**Figure 3 ijms-20-02325-f003:**
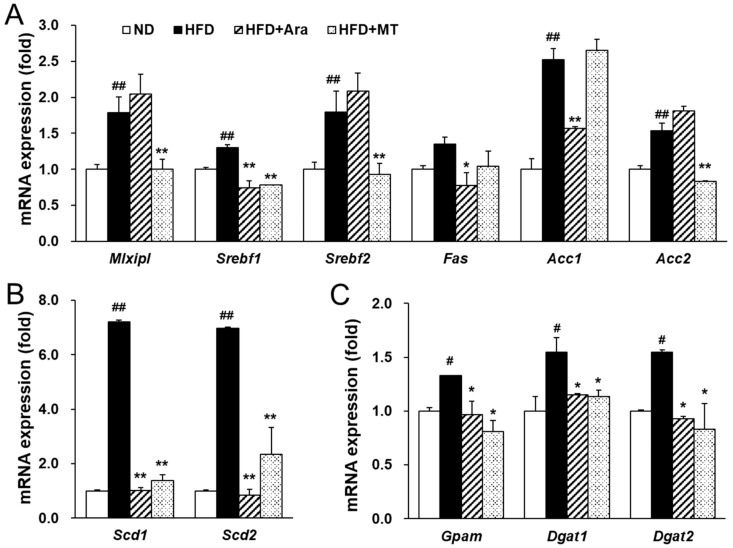
Arazyme reduced lipogenesis-related gene and protein expression in the liver. MLX interacting protein like (*Mlxipl*), sterol regulatory element-binding transcription protein 1 (*Srebf1*), fatty acid synthase (*Fas*), acetyl-CoA carboxylase 1 (*Acc1*), stearoyl-CoA desaturase-1 (*Scd1*), glycerol-3-phosphate acyltransferase (*Gpam*), diacylglycerol *O*-acyltransferase 1 (*Dgat1*), peroxisome proliferator-activated receptor alpha (*Ppara*), acyl-coenzyme A oxidase (*Acox*), carnitine palmitoyltransferase alpha (*Cpt1a*), and uncoupling protein 2 (*Ucp2*), glyceraldehyde-3-phosphate dehydrogenase (*Gapdh*). (**A**–**D**) The mRNA expression levels of lipogenesis and lipid oxidation-related genes in the liver were detected by real-time qRT-PCR. (**E**,**F**) Hepatic SREBP-1 and FAS protein expression was detected by Western blotting with specific antibodies. Data are presented as means ± SE (*n* = 4). ^#^
*p* < 0.05, ^##^
*p* < 0.01 vs. ND; * *p* < 0.05, ** *p* < 0.01 vs. HFD by one-way ANOVA.

**Figure 4 ijms-20-02325-f004:**
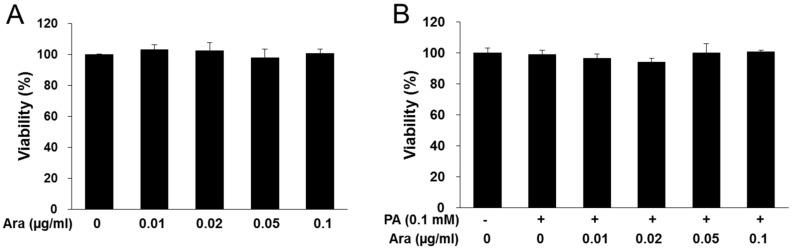
Arazyme reduced intracellular lipid accumulation in PA-induced HepG2 cells. (**A**,**B**) Viability of arazyme in HepG2 cells treated with arazyme. Cells were incubated with the indicated concentrations of arazyme for 24 h without (**A**) or with (**B**) PA. Oil-red O staining images (**C**, × 200 magnification) and intracellular TG contents (**D**). (**E**,**F**) Protein expression levels of SREBP-1 and FAS were detected by Western blotting. Data are presented as means ± SD (*n* = 3). ^#^
*p* < 0.05, ^##^
*p* < 0.01 vs. cells treated with medium only; * *p* < 0.05, ** *p* < 0.01 vs. cells treated with PA only by Student’s *t*-tests.

**Figure 5 ijms-20-02325-f005:**
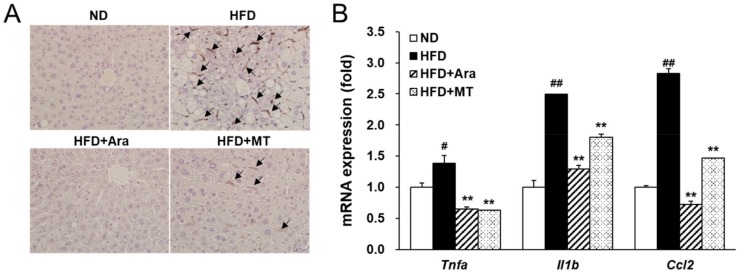
Arazyme suppressed steatohepatitis in HFD-fed NAFLD-like mice. (**A**) Liver sections immunohistochemically stained for F4/80 (×400 magnification). F4/80-positive macrophages were marked by black arrows. (**B**) The mRNA expression levels of inflammation-related genes in the liver were detected by real-time qRT-PCR. Data are presented as means ± SE (*n* = 4). ^#^
*p* < 0.05, ^##^
*p* < 0.01 vs. ND; ** *p* < 0.01 vs. HFD by one-way ANOVA. Tumor necrosis factor α (*Tnfa*), interleukin-1β (*Il1b*), and chemokine ligand 2 (*Ccl2*).

**Figure 6 ijms-20-02325-f006:**
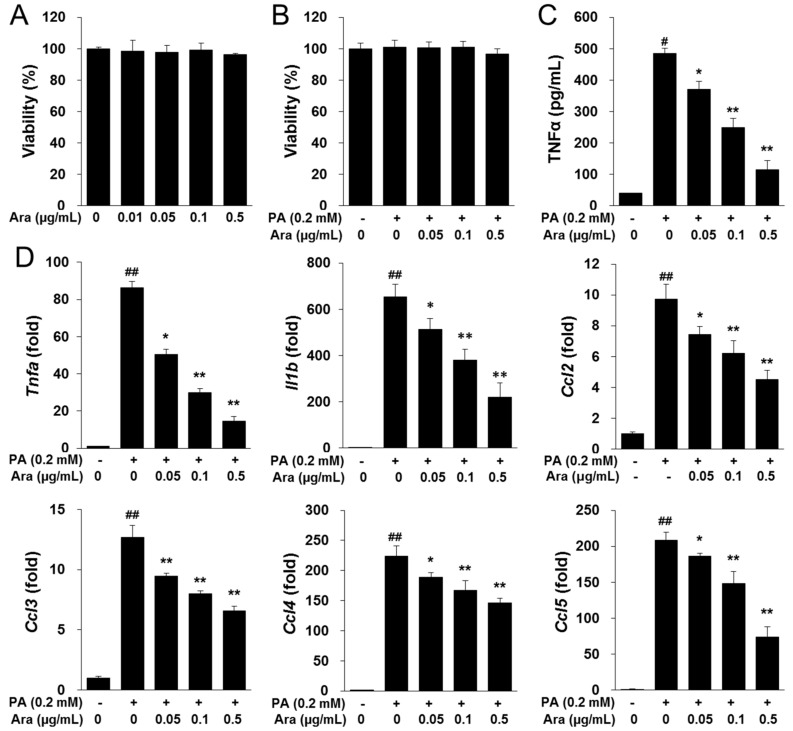
Arazyme reduced PA-induced inflammation in RAW264.7 macrophages. (**A**,**B**) Viability of arazyme in RAW264.7 cells. Cells were incubated with the indicated concentrations of arazyme for 2 h without (**A**) or with (**B**) PA. The concentration of TNFα secretion (**C**) and mRNA expression (**D**) of arazyme in PA-induced RAW264.7 macrophages. Data are presented as means ± SD (*n* = 3). ^#^
*p* < 0.05, ^##^
*p* < 0.01 vs. cells treated with media only; * *p* < 0.05, ** *p* < 0.01 vs. cells treated with PA only by Student’s *t*-tests.

**Table 1 ijms-20-02325-t001:** Body and organ weights in high-fat diet (HFD)-fed non-alcoholic fatty liver disease (NAFLD)-like mice *^a^*.

	ND	HFD	HFD+Ara	HFD+MT
Initial body weight (g)	23.5 ± 0.3	23.6 ± 0.3	23.3 ± 0.4	23.5 ± 0.4
Final body weight (g)	38.8 ± 0.8 ^‡^	44.9 ± 1.4 ^‡, ##^	44.4 ± 0.8 ^‡^	43.6 ± 1.9 ^‡^
Weight gain (g)	15.5 ± 0.5	21.3 ± 1.2 ^##^	21.5 ± 0.7	20.2 ± 1.5
Food intake (g/day)	2.3 ± 0.1	2.0 ± 0.2 ^#^	2.0 ± 0.1	2.1 ± 0.1
Liver (g)	1.28 ± 0.04	1.36 ± 0.09	1.29 ± 0.06	1.35 ± 0.13

*^a^* Values are presented as means ± SE (*n* = 10). ^‡^
*p* < 0.01 vs. initial body weight in each group; ^#^
*p* < 0.05, ^##^
*p* < 0.01 vs. HFD by one-way ANOVA. Normal diet (ND), arazyme (Ara), milk thistle (MT).

**Table 2 ijms-20-02325-t002:** Plasma metabolic factor measurements in HFD-fed NAFLD-like mice *^a^*.

	ND	HFD	HFD+Ara	HFD+MT
Glucose				
Initial (mg/dL)	99.9 ± 3.5	100.6 ± 3.7	99.5 ± 2.6	100.9 ± 4.3
Final (mg/dL)	126.4 ± 10.3 ^†^	169.0 ± 12.9 ^‡, #^	145.8 ± 7.9 ^‡^	152.5 ± 14.7 ^‡^
Insulin				
Initial (ng/mL)	0.5 ± 0.1	0.5 ± 0.1	0.5 ± 0.1	0.5 ± 0.1
Final (ng/mL)	0.9 ± 0.2 ^‡^	2.3 ± 0.5 ^‡, #^	2.5 ± 0.4 ^‡^	2.3 ± 0.4 ^‡^
HbA1c (%)	5.05 ± 0.09	5.41 ± 0.12 ^#^	5.15 ± 0.06	5.12 ± 0.13 *
HOMA-IR	6.9 ± 1.3	19.1 ± 4.7 ^#^	16.4 ± 3.0	17.5 ± 4.3
TG (mg/dL)	118.2 ± 7.5	142.7 ± 9.5 ^#^	118.4 ± 4.2 *	111.7 ± 10.0 **
TC (mg/dL)	144.5 ± 6.4	183.1 ± 9.4 ^#^	170.6 ± 7.2	162.0 ± 13.0
HDL-C (mg/dL)	74.4 ± 6.0	73.2 ± 2.6	68.0 ± 2.2	63.7 ± 3.6
HDL-C/TC (%)	50.7 ± 3.5	40.9 ± 2.0 ^##^	39.7 ± 1.5	40.4 ± 1.6
NEFA (mEq/L)	2.1 ± 0.1	2.7 ± 0.1 ^#^	2.3 ± 0.1 *	2.2 ± 0.2 *
AST (IU/L)	96.5 ± 6.1	117.4 ± 5.8 ^#^	105.8 ± 11.3	111.8 ± 10.4
ALT (IU/L)	26.0 ± 3.6	45.3 ± 10.2 ^#^	41.5 ± 5.7	40.2 ± 4.2

*^a^*Values are presented as means ± SE (*n* = 10). ^†^
*p* < 0.05, ^‡^
*p* < 0.01 vs. initial levels in each group; ^#^
*p* < 0.05, ^##^
*p* < 0.01 vs. ND; * *p* < 0.05, ** *p* < 0.01 vs. HFD by one-way ANOVA. Glycated hemoglobin (HbA1c), homeostatic model assessment for insulin resistance (HOMA-IR), triglyceride (TG), total cholesterol (TC), high-density lipoprotein cholesterol (HDL-C), non-esterified fatty acid (NEFA), aspartate transaminase (AST) and alanine transaminase (ALT).

**Table 3 ijms-20-02325-t003:** Sequences of primers used for real-time qRT-PCR.

Gene (Number)	Forward Primer	Reverse Primer
*Acc1* (NM_133360)	AGTTTCCCAGCCAGCAGATT	ATCCATCACCACAGCCTTCA
*Acc2* (NM_133904)	CCCATCACCACTCCTTCTGA	GTCCGAGTCTCCACAGCAAT
*Acox* (NM_015729)	GCTGGGCTGAAGGCTTTTACTA	AATCCCACTGCTGTGAGAATAGC
*Ccl2* (NM_011333)	TCACCTGCTGCTACTCATTC	TACAGAAGTGCTTGAGGTGG
*Ccl3* (NM_011337)	CACCCTCTGTCACCTGCTCAA	TGGCGCTGAGAAGACTTGGT
*Ccl4* (NM_013652)	CTAACCCCGAGCAACACCAT	AGCCCATTGGTGCTGAGAAC
*Ccl5* (NM_013653)	TCCCTGTCATTGCTTGCTCTAG	GAGCAGCTGAGATGCCCATT
*Cpt1α* (NM_013495)	CTGCACTCCTGGAAGAAGAA	GTTCTTCGTCTGGCTTGACA
*Dgat1* (NM_010046)	ACAACCTGACCTACCGAGAT	AGTAGGGACCATCCACTGTT
*Dgat2* (NM_026384)	GCTGGCATTTGACTGGAACA	TGGTCAGCAGGTTGTGTGTCTT
*Fas* (NM_007988)	TGTGAGTGGTTCAGAGGCAT	TTCTGTAGTGCCAGCAAGCT
*Gapdh* (NM_001001303)	ACATCATCCCTGCATCCACT	AGATCCACGACGGACACATT
*Gpam* (NM_001356285)	CCGGAAGAGGCCCTTCGTGG	GCCAGCCATCCTCTGTGCCT
*Il1b* (NM_008361)	ATGGCAACTGTTCCTGAACTCAACT	ATATTCTGTCCATTGAGGTGGAGAGCT
*Mlxipl* (NM_021455)	CAGATGCGGGACATGTTTGA	AATAAAGGTCGGATGAGGATGCT
*Pparα* (NM_011144)	CCTGAACATCGAGTGTCGAA	GTACTGGCATTTGTTCCGGT
*Scd1* (NM_009127)	ACGCCGACCCTCACAATTC	AGTTTTCCGCCCTTCTCTTTG
*Scd2* (NM_009128)	CCGTGGCTTCTTTTTCTCTCA	TTCCGCCCTTCTCTTTGACA
*Srebf1* (NM_011480)	GAGCGAGCGTTGAACTGTAT	ATGCTGGAGCTGACAGAGAA
*Srebf2* (NM_033218)	TCCTCCATCAACGACAAAATCA	ACTTGTGCATCTTGGCATCTGT
*Tnfa* (NM_001278601)	CTCAGATCATCTTCTCAAAATTCGAGTGACA	CTTCACAGAGCAATGACTCCAAAGT
*Ucp2* (NM_011671)	GCCTCTACGACTCTGTCAAA	CTTCGACAGTGCTCTGGTAT

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
