# Peer review of "Arazyme Suppresses Hepatic Steatosis and Steatohepatitis in Diet-Induced Non-Alcoholic Fatty Liver Disease-Like Mouse Model"

_ijms, 2019, doi:10.3390/ijms20092325_

Round 1

Reviewer 1 Report

Manuscript Review Comments

Title: “Arazyme suppresses hepatic steatosis and steatohepatitis in diet-induced non-alcoholic fatty liver disease-like mouse model” (ijms-481178)

The authors present a manuscript in which they perform a study on the effects of arazyme in liver steatosis. They perform a study on rodents with high fat diet and also perform an in vitro study on human liver cells. They conclude that supplementation of arazyme may have potential hepatoprotective effects in NAFLD. Although it is a comprehensive study, this reviewer has found several shortcomings in the manuscript that should be addressed.

Comment to the authors

General comments

-          The presented manuscript present overlapping with a previous publication from the same authors (Li H, Kim UH, Yoon JH, Ji HS, Park HM, Park HY, et al. Black Soybean Leaf Extract Suppresses Hyperglycemia and Hepatic Steatosis by Enhancing Adiponectin Receptor Signaling and AMPK Activation. Journal of Agricultural and Food Chemistry. 2018;67(1):90-101). This overlapping is fundamentally present on material & methods and also de figure footnotes. I recommend to rewrite these sections in order to reduce this overlapping and consider citing previous manuscripts, to avoid potential plagiarism issues.

-          The manuscript needs and English grammar and style revision.

-          Consider performing separate manuscripts. Since material & methods and results section are too big and maybe separate studies could beneficiate o better introduction and discussion section, as well as more descriptive methodology.

Specific comments:

Title page and abstract:

-          Avoid the use of acronyms in the abstract for better understanding (also the acronym for TNFα is not correctly spelled).

-          There is no section in the abstract related to materials & methods.

-          Use keywords that are MeSH terms for better indexation of the manuscript.

Introduction

-          I think that introduction section would benefit from a more extended comment on oral health, maybe making more emphasize on previous evidence on the effects of eating disorders on periodontal health.

-          Several sentences of introduction do not have any citation that supports the statement. Please revise introduction (beginning of page 2) and add citations to support the statements.

-          Line 50: state the acronym of FAO.

-          The last sentence of introduction is not an objective but a result. If there were other objectives apart from the already stated, they should be added accordingly.

Results and discussion

-          Table 1: why not presenting a variable with the mean difference in weight in each group? I think table 1 could present better the results. Also, present the exact p-values in a separate column and state the statistical test used in table footnotes.

-           Table 2: Same comments than the ones of table 1. The effect of supplementation cannot be evaluated without knowing the baseline values, making a within-group analysis and a between group analysis (I interpret that these are the final values). Again. Please include the exact p-values in the table since it is a small and concise table. Also detail the statistical tests used in table footnotes (b is not described in the footnotes).

-          Figure 2: Authors should present the same section for H&E, IHC and Oil Red O images in order to compare the same sections.

-          Authors present a quite short discussion section considering the large results that they present.

-          Last paragraph of discussion has a different line spacing.

Materials and Methods

-          Please comment on the study design and follow an appropriate guideline for the manuscript preparation and specify this in the methods section. In this case, ARRIVE guidelines are the appropriate for animal studies.

-          Please check for the font size of some parts of material & methods.

-          Oil red O it is not an IHC technique but a staining technique. IHC necessarily needs the use of antibodies for marker detection. If that was de case, IHC technique needs to be described in more detail (antigen unmasking, antibody used and commercial reference, reagents for staining of the antibody etc.).

-          Make sure all the commercial references are complete (e.g. line 315: Roche Diagnostics; line 341: Sigma-Aldrich).

-          As stated before, the experiment performed on cell in vitro do not relate to the results obtained in animals. It is separate experiment and therefore included in a different study or as a separate part with its own objective and a section in introduction with the rationale for the use of these cells.

-          Do the authors aim to test for the potential toxicity of arazyme in the studied cells?

Author Response

Dear reviewer,

  Thank you for your effort in reviewing our paper and for the valuable comments that have helped us to improve the manuscript. We agree with your comments and, therefore, we have revised our manuscript accordingly, as indicated in our responses to your comments listed in atteched file.

Reviewer 2 Report

In this study, Li et al. seek therapeutic effects of arazyme using high fat diet model. Data sound solid and I have only minor comments for this manuscript.

1.      Tables and figures are confusing with a, b, and ab. They need to represent clearly that differences are significant compared to normal diet or high fat diet. Please edit them.

2.      In Figure 1A, please perform statistical analyses to compare groups using two-way ANOVA with post-hoc analysis or similar.

3.      Table 2 shows TG and TC in mg/dL and Figure 2C and D show in mg/g. Data for TC in Figure 2D do not look identical to those shown in Table 2. Why is that? Different data using different results? Please describe clearly and make it simple. Probably Figure 2C and D are not needed (if data are same as those in Table 2) and Figure 1A and B can be merged with Figure 2A and B.

4.      Error bars are missing in some data (eg., Figure 3 ND group, Figure 4F, Figure 6C etc). All data should show error bars. Sample numbers (n=…) are also missing in some data. Please show error bars and sample numbers clearly in all data.

5.      Please show arrows in histological images to point out area of interest (F4/80+ cells) in Figure 5A.

Author Response

Dear reviewer,

Thank you for your effort in reviewing our paper and for the valuable comments that have helped us to improve the manuscript. Our responses to the comments are provided in atteched file.

Round 2

Reviewer 1 Report

My previous comments have been addressed properly and the manuscript quality has greatly improved. I can be acceptable for publication in its present form.